# Up for the challenge: Power motive congruence drives nurses to craft their jobs and experience well-being

Rawan Ghazzawi[1,2,3], Athanasios Chasiotis[4], Michael Bender[2,5], Lina Daouk-Öyry[6], Nicola Baumann[7] *

1 Human Resource Studies Department, Tilburg University, Tilburg, The Netherlands, 2 Department of Social Psychology, Tilburg University, Tilburg, The Netherlands, 3 Evidence-based Healthcare Management Unit, American University of Beirut, Beirut, Lebanon, 4 Department of Developmental Psychology, Tilburg University, Tilburg, The Netherlands, 5 Gratia Christian College, Hong Kong, PR China, 6 Department of Leadership and Organizational Behavior, BI Norwegian School of Business, Oslo Campus, Oslo, Norway, 7 Department I–Psychology, Differential Psychology, Universität Trier, Trier, Germany

* nicola.baumann@uni-trier.de

**Data Availability Statement:** The data files are available from PsychArchives via the link below: https://doi.org/10.23668/psycharchives.12923.

**Funding:** NB: The publication was supported by the Open Access Fund of Universität Trier The funders

## Abstract

Job crafting is the behavior that employees engage in to create personally better fitting work environments, for example, by increasing challenging job demands. To better understand the driving forces behind employees' engagement in job crafting, we investigated implicit and explicit power motives. While implicit motives tend to operate at the unconscious, explicit motives operate at the unconscious level. We focused on power motives, as power is an agentic motive characterized by the need to influence your environment. Although power is relevant to job crafting in its entirety, in this study, we link it to increasing challenging job demands due to its relevance to job control, which falls under the umbrella of power. Using a cross-sectional design, we collected survey data from a sample of Lebanese nurses (N = 360) working in 18 different hospitals across the country. In both implicit and explicit power motive measures, we focused on *integrative power* that enable people to stay calm and integrate opposition. The results showed that *explicit power* predicted job crafting (H1) and that *implicit power* amplified this effect (H2). Furthermore, job crafting mediated the relationship between congruently high power motives and positive work-related outcomes (H3) that were interrelated (H4). Our findings unravel the driving forces behind one of the most important dimensions of job crafting and extend the benefits of motive congruence to work-related outcomes.

## Introduction

Job crafting is a set of creative work behaviors that employees engage in to create a better fit between their needs and preferences and their jobs [1]. Among other work-related concepts, job crafting is well aligned with the recent trend to focus on proactive work behaviors that allow employees to create work environments they engage in productively [2]. Many findings

had no role in study design, data collection and analysis, decision to publish, or preparation of the manuscript.

**Competing interests:** The authors have declared that no competing interests exist.

show that job crafting has indeed positive effects on work-related outcomes such as work engagement, job satisfaction, and work performance [3], however, we still do not know a lot about the personal antecedents of job crafting in general [4] and among nurses specific. This can create a level of unclarity as to who is more likely to engage in job crafting. Addressing this gap is essential as job crafting is a personal and proactive work behavior driven by basic needs [2, 5]. Therefore, it is important to understand the personal characteristics that drive employees' engagement in job crafting.

Studies on personal antecedents of job crafting have mainly focused on traits such as the big five and self-efficacy [3] and basic needs at the explicit (intrinsic and extrinsic) level [6, 7], which were restricted to self-report. In contrast, implicit (unconscious) basic motives for power, achievement, and affiliation that are driving forces of human behavior [8, 9] have been neglected thus far. Motives, however, are dually represented at explicit and implicit levels [10]. Whereas explicit motives are well captured by self-report, implicit motives require projective techniques because they are not consciously accessible to individuals [11]. Implicit and explicit motives function independently but may collaborate in driving people to craft their jobs for the better. Congruently high implicit and explicit power motives, for example, have been found to promote spontaneous helping [12], sustained volunteering [13], and personal well-being [14–16]. Therefore, we investigated the effects of both implicit and explicit power motives on job crafting and occupational well-being.

Despite the large amount of research on job crafting, few studies have explored it in the context of nursing. Moreover, a very limited number of these studies investigated the personal antecedents of job crafting in the nursing context [e.g., 17]. Knowing that the well-being of health care professionals is an important predictor of patient quality of care [18, 19] and individual and organizational performance [20, 21] it would be of high value to investigate the influence that job crafting has on important positive work outcomes in such a professional context. Therefore, we investigated the effects of job crafting on work engagement and job satisfaction–two of the strongest positive work outcomes of job crafting [3]. In the following, we highlight an important dimension of job crafting, elaborate on the power motives assumed to drive job crafting, and propose an indirect path from power motive congruence through job crafting to positive work outcomes (Fig 1A).

## Job crafting as increasing challenging job demands

What do employees actually do when crafting their jobs? Adopting the job demands-resources theory [JD-R; 22], Tims and Bakker [1] defined job crafting in terms of shaping job demands and job resources. The corresponding scale differentiates increasing challenging job demands, decreasing hindering job demands, and increasing social and structural job resources [23]. In the present study, we focused on increasing challenging job demands as the relation between job demands and employee control at work is strongly supported by theory [24, 25]. Furthermore, increasing challenges has been shown to predict the most variance in the overall job crafting construct [$R^2 = 0.657$; 3]. Finally, recent literature has indicated the need for developing a deeper understanding of job crafting [4], which can be done by investigating links with specific job crafting dimensions instead of job crafting as a whole.

Increasing challenging job demands revolves around expanding one's job by taking on more tasks [1]. Examples of this behavior are volunteering to take on new projects and doing extra tasks at work. In the nursing context, this can be volunteering to help at the emergency department when there is a lot of pressure or creating a small support group for junior nurses. Individuals who engage in increasing challenges focus on positive outcomes, are eager to expand their abilities, and know that they can handle more workload [26]. Such an orientation

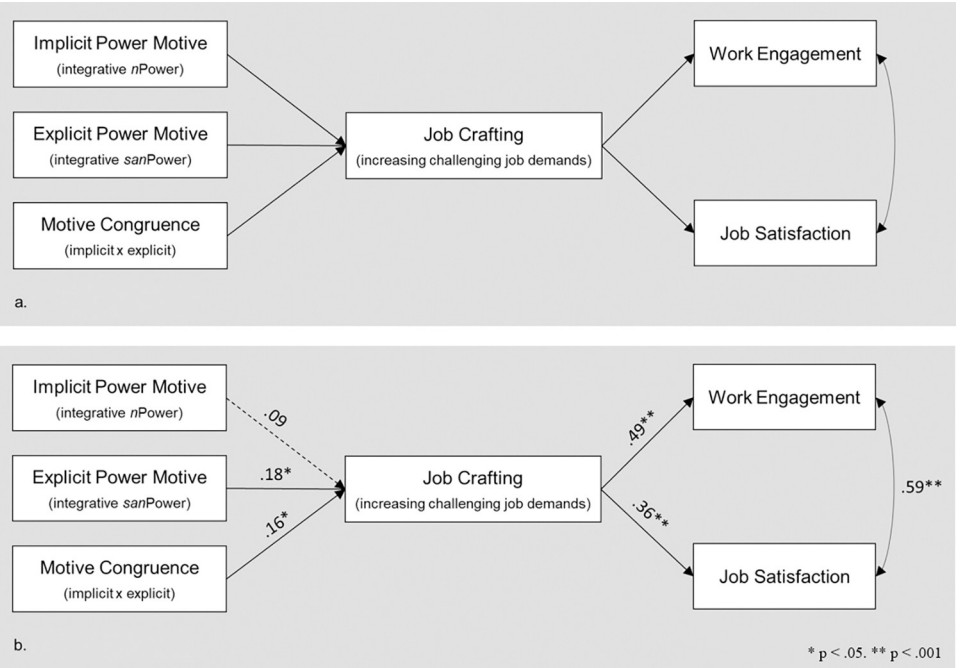

**Fig 1. Theoretical and empirical models.** The dashed arrow indicates a non-significant pathway.

requires a considerable amount of self-regulation, which is arguably required for engaging in job crafting as a whole [27]. From a motivational perspective, increasing challenging job demands can be regarded as a self-regulated approach tendency and could be driven by self-regulated ways of enacting own motives [6, 28]. Therefore, we did not only consider motive contents but also motive enactment strategies.

## Motive dispositions: What people need and how they strive

Motive research has established that implicit and explicit motives are independent sources of human motivation that differ in terms of their origins, the cues they are sensitive to, and the behavior they predict [10, 29, 30]. Implicit motives are shaped by affective experiences, are sensitive to activity-inherent incentives, and predict spontaneous behavior that people do on their own accord as well as behavioral long term trends. Explicit motives are shaped by reflection of social norms and expectations, are sensitive to social-evaluative cues, and predict planned behavior [10, 12, 31]. Since job crafting behaviors are often multifaceted and complex [1], they are likely to be influenced by both implicit and explicit motives.

Motives tell us *what* people are striving for [8, 32, 33]. The power motive is defined as striving for having impact on others (e.g., prosocial guidance, integrative leadership, dominating others), the achievement motive as striving for meeting standards of excellence (e.g., learning something, mastering challenging tasks, outperforming others), and the affiliation motive as striving for satisfying relationships (e.g., intimacy, socializing with friends, security). Nursing can be regarded as a power-profession because it is about influencing others in a prosocial manner (helping, healing, and taking care of others) [9, 33–35]. Although nurses' decisions to join the profession are driven by different motives, most of them express that they chose this profession driven by a calling to help and care for others [36]. Increasing challenges in these power-related job demands is tantamount to striving for more impact on others. Furthermore, increasing challenging job demands in any kind of profession requires asserting impact on

colleagues, superiors, and the environment as it involves volunteering on projects with others or taking over tsks from their managers [1]. This makes the power motive the most relevant driving force for job crafting.

In addition to the content of people's strivings, motive researchers have started to differentiate *how* individuals strive for what they need [37, 38]. Do they self-regulate the satisfaction of own needs or rely on the odds of circumstances? Self-regulated motive enactment is indicated by flexibility, creativity, and integrative capacities—hence the label 'integrative'–and distinct from not self-regulated (e.g., intuitive, controlled, and anxious) ways of motive enactment [12, 13, 16, 28, 39–41]. The integrative power motive is enacted by asserting one's self, making decisions, expressing emotions, staying calm in the face of conflict, and understanding and properly dealing with negative emotions [32]. Both implicit and explicit power motives can be enacted in integrative ways and may drive employees' tendency to engage in a self-regulated behavior like job crafting.

## Explicit power and job crafting

Job crafting is a planned type of coping behavior [42] exhibited in organizational settings that are bordered by external incentives and expectations. This indicates that job crafting is predicted by the explicit power motive. Explicit motives drive behavior that is planned and highly dependent on and sensitive to social-evaluative incentives [10]. Especially the self-regulated way of enacting the explicit power motive (integrative *explicit power*) makes individuals less hesitant to act, allows them to acknowledge the personal difficulties they are facing, and promotes coping in a calm, creative, and integrative manner [29, 43]. These characteristics of integrative *explicit power* resonate very well with the nature of job crafting.

Wrzesniewski and Dutton [5] argue that job crafting starts at the cognitive level and takes place in the workplace where performance indicators serve as expectations to be met. Zhang and Parker [44] have argued that cognitive crafting is relevant in driving job crafting behaviors. Moreover, a recent study by Costantini [45] indicted that employees who cognitively reframed their work engaged in more role expansion. Employees engage in job crafting based on intentions to do so [46], meaning job crafting is more planned, less spontaneous, and driven by a "reason to" motivation [47]. Accordingly, we hypothesized that integrative enactment of the explicit power motive would be positively related to job crafting.

*H1*: *Explicit power is positively related to job crafting.*

## Implicit power and job crafting

Although job crafting is a planned behavior [1, 5] that is predicted by high *explicit power*, it can be further energized by high *implicit power* [48, 49]. Notably, such a motive congruence is not always the case. As implicit and explicit motives are independent and show negligible correlations [e.g., 50], roughly half of the population exhibits motive *in*congruence. For example, they strive for having impact on others without feeling any pleasure (high *explicit power*/low *implicit power*) or miss out on opportunities for asserting oneself that would elicit positive affect [low sanPower/high nPower; 28].

Research shows that motive *in*congruence is associated with negative psychological outcomes [51–53]. Motive congruence, in contrast, has been associated with well-being and life satisfaction across motive domains and across cultures [e.g., 14, 54, 55], for a recent overview, see Chasiotis, Hofer and Bender [56]. In power-related occupations (e.g., teachers, managers, psychologists) where having impact on others is an essential job characteristic, it is especially the high/high combination of implicit and explicit power motives that has been associated

with positive outcomes [e.g., 15, 16]. Accordingly, in our sample of nurses, we hypothesized that congruently high power motives would be associated with higher job crafting.

*H2*: *Implicit power amplifies the effect of explicit power on job crafting.*

## Motive congruence, job crafting and work-related outcomes

Extensive research supports the relationship between job crafting and work engagement in general [e.g., 2, 27] and among nurses [57]. According to the job demands-resources model, motivated employees engage in job crafting and as result have more resources and experience higher levels of motivation [58]. This is referred to as the "gain spiral" that employees create for themselves to increase their work engagement via job crafting [59]. However, little is known about the motivational antecedent in this relationship. This is surprising, especially since work engagement is often referred to as a motivational process in the job demands-resources model [22]. According to self-determination theory [60–62], individuals engage in activities they find interesting and challenging enough and as a result experience positive outcomes. Thus, the end result that the individual seeks to achieve would not be the behavior, but the state that this behavior would put them in, which we argue is work engagement.

Motive congruence allows individuals to engage in behaviors that are aligned with their needs and preferences [63] and, thus, promote work engagement. Accordingly, we expected to find an indirect path from motive congruence through job crafting to work engagement. Meta-analyses indicated that challenging job demands were also positively related to job satisfaction [3], Podsakoff, LePine and LePine [64]. Therefore, we hypothesized to find a similar path from motive congruence through job crafting to job satisfaction (Fig 1A).

*H3a*: *Motive congruence, through job crafting, is positively related to work engagement.*

*H3b*: *Motive congruence, through job crafting, is positively related to job satisfaction.*

Work engagement and job satisfaction are empirically related [e.g., 65]. Therefore, we expected to find a positive relationship between work engagement and job satisfaction (Fig 1A).

*H4*: *Work engagement and job satisfaction are positively related.*

## Method

### Sample and procedure

We collected data from 482 nurses from 18 hospitals across Lebanon between March 21, 2018 and March 20, 2019 after obtaining ethical approval from the Institutional Review Board (IRB) under The Human Research Protection Program (HRPP) and The Ethics Review Board (ERB). This research was not preregistered in an independent, institutional registry. Hardcopies of the questionnaires were distributed among the nurses in each of the participating hospitals and the nurses were requested to return the filled questionnaires. At the beginning of the study the participants had the chance to read about the aim of the study and provide their written consent if applicable. Questionnaires were administered in Arabic after careful scale adaptation processes [66, 67]. A bilingual research team translated and back-translated the scales. A university professor in Arabic reviewed the items, and a committee of bilingual researchers finalized the scales. Cognitive interviews with a small group of nurses assessed item clarity. Participants who took part in the study were enrolled in a draw for monetary prizes (two of $25 and one of $50 per hospital). Moreover, one US Dollar per survey was donated to the Children's Cancer Center of Lebanon (CCCL). Of the 500 surveys that we distributed, 482 were sent back (response rate: 96.40%). The responses per hospital ranged between 6 and 47

**Table 1. Demographic characteristics.**

| | Valid Values | |
|---|---|---|
| **Characteristic** | **n** | **%** |
| Gender (Missing = 7) | | |
| Male | 73 | 20.68 |
| Female | 280 | 79.32 |
| Age (M = 32.01, SD = 8.17, Missing = 69) | | |
| < 20 | 4 | 1.37 |
| 20–30 | 144 | 49.48 |
| 31–40 | 102 | 35.05 |
| > 40 | 41 | 14.09 |
| Position (Missing = 14) | | |
| Registered Nurse | 266 | 76.88 |
| Practical Nurse | 46 | 13.29 |
| Other nursing positions | 34 | 9.83 |
| Highest Education (Missing = 8) | | |
| Diploma in Nursing | 47 | 13.35 |
| Baccalaureate[a] Technical (BT) in Nursing | 46 | 13.07 |
| Technique Supérieure[b] (TS) in Nursing | 51 | 14.49 |
| License Technique[b] (LT) in Nursing | 75 | 21.31 |
| Bachelor of Sciences (BS) in Nursing | 72 | 20.45 |
| Masters (MS) in Nursing | 43 | 12.22 |
| Other | 18 | 5.11 |
| Organizational Tenure (M = 8.66, SD = 7.28, Missing = 28) | | |
| < 1 year | 15 | 4.52 |
| 1–5 years | 122 | 36.75 |
| 5.5–10 years | 92 | 27.71 |
| 11–20 years | 76 | 22.89 |
| > 20 years | 27 | 8.13 |
| Position Tenure (M = 7.65, SD = 6.40, Missing = 26) | | |
| < 1 year | 13 | 3.89 |
| 1–5 years | 143 | 42.81 |
| 5.5–10 years | 96 | 28.74 |
| 11–20 years | 68 | 20.36 |
| > 20 years | 14 | 4.19 |

[a.] Equivalent to the last year of high school.

[b.] Technical degrees.

depending on the hospital size. The samples from the different hospitals were similar in gender, age, tenure, and education. We excluded 119 participants, since they missed answering more than three out of twelve pictures in the implicit motive test, which is the cut off number usually adopted [see 68]. To reduce bias [69], we then removed three more participants, since they had surveys with missing rates higher than 25%. The final sample size of the study was N = 360. Table 1 presents the demographic characteristics.

## Measures

**Implicit power.** We measured the implicit power motive using the Operant Motive Test [OMT; 33; see Fig 2]. The OMT consisted of 12 pictures portraying individuals in vague

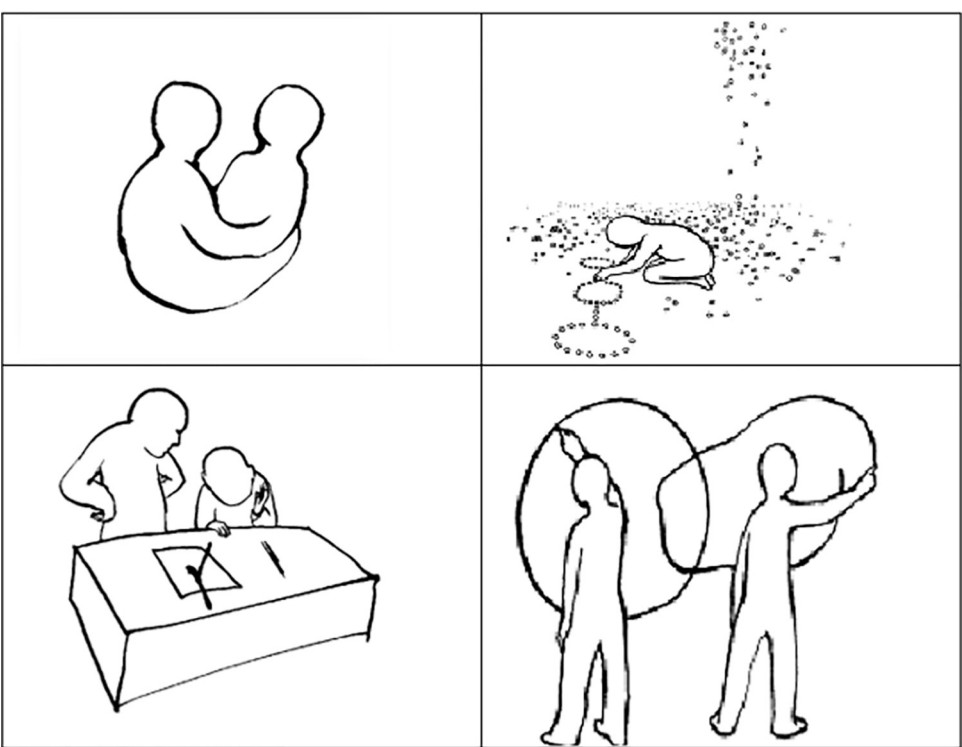

**Fig 2. Example pictures in the Operant Motive Test (OMT; Kuhl & Scheffer, 1999).** Participants were asked to select a main character and answer three open questions: *What is important for the person in this situation and what is the person doing? What does the person feel? Why does the person feel this way?* Answers were coded for motive content (power, achievement, and affiliation) and motive enactment strategies (intrinsic, intuitive, integrative, rigid, and fearful). The present study focused on the integrative enactment of the implicit power motive (integrative *implicit power*).

situations. Participants were asked to choose a main character and answer three questions related to this main protagonist. Raters coded *what* people strive for (power, achievement, affiliation) and *how* people strive: self-regulated (intrinsic, integrative) versus not self-regulated (intuitive, rigid, anxious). In the present study, we used only the self-regulated/integrative implicit power motive (integrative *implicit power*). Information about coding the OMT can be found in S2 File.

**Explicit power.**   We measured the explicit power motive using the four-item self-regulated/integrative power subscale (integrative *explicit power*) from the Motive Enhancement Test [MET; 70]. Example items are: *"I feel that most of the time I can speak my mind" "During arguments, I can often think of ways to get the other person to agree with me."* Participants rated items on a four-point Likert scale, ranging from 1 (*applies not at all*) to 4 (*applies completely*).

**Job crafting.**   We measured the job crafting dimension of *increasing challenging job demands* using a five-item subscale, that we adapted from the widely used scale by Tims, Bakker and Derks [23]. We developed this scale as part of a larger project on distinguishing instrumental and affective qualities of job crafting (see S1 File). Example items are: *"I try to understand how my work tasks are related to one another"* and *"I am always the first one to try new developments out to enhance my skills".* Participants rated items on a five-point Likert scale, ranging from 1 (*never*) to 5 (*very often*).

**Work engagement.**   We used the 17-item Utrecht Work Engagement Scale [UWES; 71, 72]. An example item is: *"At my job, I feel strong and vigorous".* Participants rated items on a seven-point Likert scale, ranging from 0 (*never*) to 6 (*every day*).

**Table 2. CFA model fit information.**

| Scale | χ2 | RMSEA | 90% CI | CFI | TLI |
|---|---|---|---|---|---|
| Job Crafting | 9.463 | .056 | .000, .110 | .988 | .976 |
| Job Satisfaction | Model Saturated | | | | |
| Work Engagement | 259.977 | .071 | .060, .082 | .956 | .943 |
| Explicit Power | 3.657 | .027 | .000, .106 | .986 | .972 |

χ2 = Chi-squared; RMSEA = Root mean square error of approximation; CFI = Comparative fit index; TLI = Tucker Lewis index.

## Job satisfaction

We measured job satisfaction using the three-item scale developed by Tims, Bakker and Derks [73]. An example item is: *"I am satisfied with my current work"*. Participants rated items on a 5-point Likert scale ranging from 1 (*totally disagree*) to 5 (*totally agree*).

# Results

## Data preparation

Before testing the hypotheses of the study, we tested the construct validity of the scales that we used. Using Confirmatory factor analysis (CFA), we checked the structure of *explicit power*, job crafting, work engagement, and job satisfaction. As listed in Table 2, we obtained satisfactory fit indices overall.

Based on the codes generated from the OMT, we used the score of the third level of the implicit power motive (integrative *implicit power*). This score is binary with "1" indicating that the participant had at least one implicit self-regulated power code and "0" that the participant had none. Descriptive statistics for the outcome variable showed that job crafting was relatively normally distributed as it was within the -0.5 and 0.5 range (skewness = -0.45). Hence, we did not transform the data. Table 3 presents descriptive information and zero-order correlations.

## Hypothesis testing

To test our hypotheses, we used structural equation modeling (SEM) in Mplus [74]. We tested the theoretical model illustrated in Fig 1A. The results indicated that the model fit the data very well ($\chi^2$ (6, $N$ = 288) = 9.013; $p$ = .173, RMSEA = .042, 90% CI: .000, .094, CFI = .988, TLI = .975, AIC = 1938.729, BIC = 1982.685). Fig 1B shows the resulting empirical model.

Consistent with H1, results revealed a significant positive direct effect of the explicit power motive (integrative *explicit power*) on job crafting ($\beta$ = .18, $p$ = .04). The direct effect from the

**Table 3. Means, standard deviations, zero-order correlations, and Cronbach's alpha coefficients (in parentheses).**

| | $M$ | $SD$ | (2) | (3) | (4) | (5) |
|---|---|---|---|---|---|---|
| (1) Implicit Power | | | .12* | .11* | .04 | .03 |
| (2) Explicit Power | 2.74 | 0.60 | (.44) | .35** | .33** | .21** |
| (3) Job Crafting | 3.79 | 0.69 | | (.79) | .47** | .32** |
| (4) Work Engagement | 4.52 | 1.14 | | | (.94) | .65** |
| (5) Job Satisfaction | 3.75 | 0.80 | | | | (.84) |

The implicit power motive is measured by the integrative facet in the Operant Motive Test (OMT) and a binary variable: 0 = not present (64.70%), 1 = present (35.30%). The explicit power motive is measured by the integrative facet in the Motive Enactment Test (MET). Job crafting is measured by the dimension of increasing challenging job demands.

\* $p$ < .05; \*\* $p$ < .001

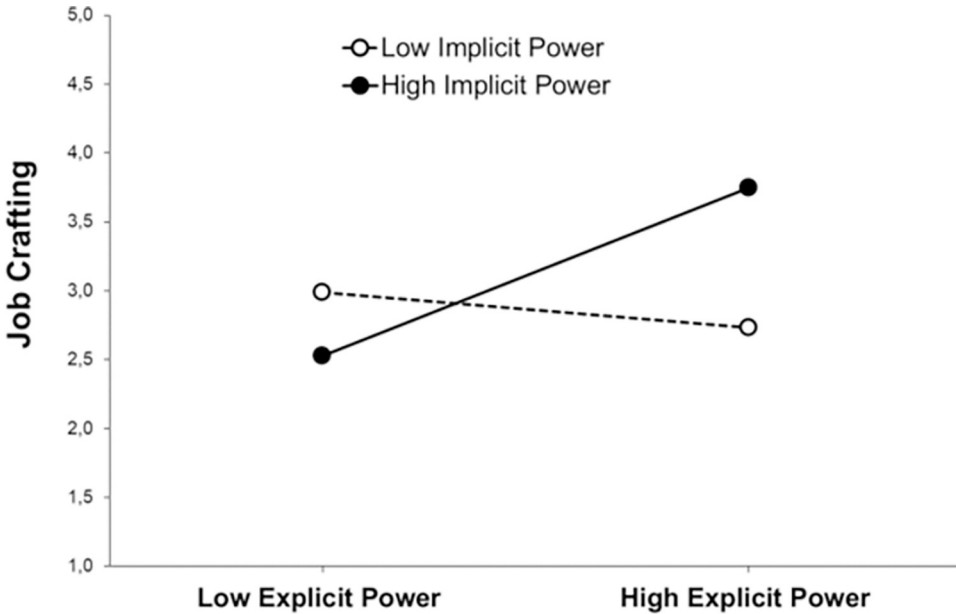

**Fig 3. Simple slope analysis.** Job crafting (i.e., increasing challenging job demands) as a function of implicit and explicit power motives (N = 360).

implicit power motive (integrative *implicit power*) on job crafting was not significant ($\beta$ = .09, $p$ = .09). Consistent with H2, there was a significant interaction effect of explicit x implicit power motives on job crafting ($\beta$ = .16, $p$ = .03). The interaction effect is illustrated in Fig 3. This indicates that there was a moderation effect. Simple slope analysis showed that, among participants with a high implicit power motive, there was a positive relationship between the explicit power motive and job crafting ($\beta$ = .61, $p$ < .001). In contrast, among participants with a low implicit power motive, there was no significant relationship between the explicit power motive and job crafting ($\beta$ = -.13, $p$ = .53). This indicates that power motive congruence promotes the engagement in job crafting.

As illustrated in Fig 1B, there was a significant relationship between job crafting and work engagement ($\beta$ = .49, $p$ < .001) as well as job crafting and job satisfaction ($\beta$ = .36, $p$ < .001). Consistent with H3a, the indirect path from motive congruence through job crafting to work engagement was significant ($\beta$ = .08, $p$ = .03). Moreover, consistent with H3b, the indirect path from motive congruence through job crafting to job satisfaction was significant ($\beta$ = .06, $p$ = .04). Finally, consistent with H4, job satisfaction and work engagement were significantly related ($\beta$ = .59, $p$ < .001). The results are discussed below.

## Discussion

Successfully completing challenging job demands can make employees feel in control and satisfy their needs. Although some employees may not be up for the challenge, others become active to even further increase challenges. In the present study, we combined theoretical approaches from motive research [8] and work behavior research [22] to enhance our understanding of this type of job crafting behavior: increasing challenging job demands. We conducted our study in the context of nursing–a power-related profession–and tested nurses' implicit and explicit power motives as driving forces behind job crafting. Even though the nursing profession is bordered by clinical standard operating procedures, researchers have argued that they can still engage in shaping their work within these boundaries [5]. This idea

has been supported by the numerous studies on job crafting conducted in the context of nursing. The concept of proactivity and non-compliance with established protocols is not unfamiliar within the nursing discipline, as these can be deliberate efforts to improve current procedures [75]. In addition, we looked at work engagement and job satisfaction as work-related outcomes of job crafting.

## Power motives drive job crafting

Our results show that nurses who score higher on the *explicit power* motive were more likely to engage in job crafting (H1). Furthermore, this effect was moderated by the *implicit power* motive indicating that congruently high power motives promote job crafting (H2). This finding is in line with motive research and the growing number of findings on positive effects of motive congruence across motives for power [16], achievement [51], and affiliation [76], across cultures [56], and across important aspects of people's personal life such as well-being [51], medication intake [76], eating behavior [77], and identity development [54]. In the present study, we empirically show that motive congruence had positive associations beyond the personal lives of individuals and influenced work behavior: Nurses with power motive congruence (i.e., high *implicit power* and high *explicit power*) did not simply endure a given work environment but actively crafted their jobs and increased challenges to a level that made them feel more engaged in their work and, in turn, more satisfied with their jobs.

## Job crafting involves self-regulation

Bakker and de Vries [78] argued that job crafting is a form of self-regulated, adaptive work behavior that can serve as a buffer against burnout. Self-regulation facilitates creative problem solving, flexibility, and coping [79]. The present study findings support the assumed relationship between self-regulation and job crafting. We did not only measure motive contents (what people strive for), but also motive enactment strategies (how people strive). We focused on the self-regulated, integrative enactment of power motives. People high in integrative *implicit power* see options for staying calm in the face of opposition, integrating different views, and supporting autonomy [33]. People high in integrative *explicit power* feel they can speak their mind and get others to agree with them [70]. The present findings show that, together, they empowered nurses to craft their jobs according to their own needs. Thus, it does not only take congruently high power motives to engage in job crafting, but also self-regulated ways of enacting both implicit and explicit power motives. The finding supports the notion of job crafting as a type of self-regulated work behavior [78].

## Work-related outcomes

Consistent with previous research, increasing challenging demands was positively related with work engagement and job satisfaction [3, 44]. This was not surprising as job crafting in general has been shown to be positively related to a wide range of positive work outcomes [80]. Moreover, a recent meta-analysis by Boehnlein and Baum [80] showed that approach crafting, including increasing challenging job demands, is positively linked to different types of employee well-being including work engagement. We extend this literature by supporting such a robust relationship in an underrepresented non-Western sample and by providing empirical support for the role of motives in increasing the level of motivation among employees (gain spiral) [58]. What was also significant were the indirect paths from motive congruence to work engagement and job satisfaction though increasing challenging job demands. There is strong and numerous evidence that allude to the positive influence of motive congruence on the well-being of individuals [69]. Building on this evidence, we extend the benefits of

motive congruence to include work-related outcomes, namely work engagement and job satis-faction. Scoring high on both implicit and explicit power motives indicates a level of self-deter-mination and emotional maturity that allows individuals to approach challenges, deal with negative emotions, and as a result experience positive work outcomes. This is in line with the expectancy theory of motivation [81] that states that (good) performance depends on how much the individual expects to receive an outcome that they consider valuable. Given that individuals who score high on integrative power motives are confident that their skills and effort will yield positive outcomes, they are likely to approach challenging work demands and receive internal rewards and gratification. In line with our findings, Bakker and Sanz-Vergel [82] previously showed that when challenging job demands are high, employees with personal resources can thrive and experience higher work engagement. Accordingly, Esteves and Lopes [83] suggested that hospitals provide training for nurses to improve their confidence in taking on challenging job demands.

## Theoretical contributions

This paper makes important theoretical contributions. First, we provide an in-depth analy-sis of the job crafting dimension of increasing challenging job demands. This is informative as recent literature has urged researchers to look at links with specific job crafting dimen-sions instead of job crafting as a whole [3]. Second, we extend our knowledge about driving forces behind job crafting to motive dispositions that have been neglected as personality antecedents so far. Thereby, we integrate two separate research fields: motive research [10] and job crafting research [1, 5]. Third, by using the Operant Motive Test (OMT) to measure implicit motives, we adhere to calls to move personality research beyond self-report (e.g., Baumeister et al., 2007) and to calls to generate more human resource management-centric literature on projective tests [84]. Finally, we extend the well documented benefits of motive congruence [8, 12] to include work-related outcomes, namely work engagement and job satisfaction.

## Practical implications

The findings of this study have downstream consequences for the selection and training of nurses in hospitals. Our results indicate that nurses who score high on both implicit and explicit power motives and who engage in job crafting, experience high levels of job satisfac-tion and work engagement. Nurses' job satisfaction and work engagement have been shown to strongly predict inpatient satisfaction and quality of care respectively [85, 86]. This is particu-larly important for hospitals in the Middle East as they suffer from less ideal work conditions and a lack of valid recruitment and retention practices [87]. Hospitals may employ or consult trained psychologists to assess nurses' motives and place nurses at positions fitting their profile.

Furthermore, hospitals could invest into training programs to develop motive congru-ence and integrative competencies among their nurses. Roch, Rösch and Schultheiss [88], for example, show that a congruence-enhancement training of 3.5 hours significantly increased power motive congruence (as assessed three weeks before and six weeks after the training); treatment-based increases in power motive congruence were associated with increases in well-being. Furthermore, Baumann and Kuhl [28] show that a multi-faceted three-hour resilience training (Study 3) as well as a more specific 30-minute self-regulation exercise (Studies 4 and 5) significantly increased integrative motive enactment. The present findings suggest that both motive congruence and integrative motive enactment are worthy targets for training.

## Limitations and future research directions

This study has some limitations that should be acknowledged. First, the scale that we used to measure the integrative explicit power motive had a low reliability in our sample, although it showed an adequate structure in our CFA. Some factors related to the non-WEIRD (Western, Educated, Industrialized, Rich, and Democratic) context of the study and the characteristics of the scale itself could have contributed to this score [89]. We elaborate on these aspects in the S1 File. Future research should test the equivalence of this scale across samples that differ in socio-cultural characteristics. Second, our study is cross-sectional and does not allow us to draw causal inferences. Although we view power motive congruence as a driving force behind nurses' engagement in job crafting, it might also be the case that engagement in job crafting indicates a self-regulatory ability that promotes motive congruence [63, 90]. Future research should adopt a longitudinal design to investigate the relationships in our study over time to make more informed conclusions.

Third, we focused on implicit and explicit power motives and their interplay as antecedents of job crafting. While some employees may provide service due to the self-regulated enactment of power motives, other might engage in job crafting for social reasons (i.e., affiliation motive) or for career reasons (i.e., achievement motive). The examination of such interaction effects for other motive domains, however, has not been realized in the present research, as self-regulated power motives were assumed to constitute the most relevant motives. Future research on job crafting would also profit from addressing the interplay between self-reported motives and their implicit counterparts with respect to other dimensions of job crafting [see also 40].

Finally, in line with constraints on generality statements [91], we like to point out that the study was conducted within a specific socio-cultural and professional context: the nursing profession in Lebanon. Accordingly, future research taking its results into consideration must acknowledge the lack of evidence on the generalizability of the study's results. More research is needed in order to test the robustness and validity of our conclusions across different contexts and samples.

## Conclusion

The present study shows that it takes power motives to be up for more challenge at work. However, a strong explicit, self-attributed power motive is not the whole story. Employees' *explicit power* motive has to be conjoined by a high *implicit power* motive and both have to be enacted in a self-regulated, integrative manner. Our findings support power motive congruence (i.e., high integrative *implicit power* and high integrative *explicit power*) as a driving force behind job crafting in the nursing context. Furthermore, taking on challenging job demands may allow nurses to feel more engaged at work and more satisfied with their jobs. Future studies may elaborate the causal role of power motives in job crafting by training motive congruence and/or integrative motive enactment. In the nursing context, this could benefit the work well-being of the nurses, which can also enhance patient care.

## Supporting information

**S1 File. Development and validation of the adapted job crafting scale.**
(DOCX)

**S2 File. Coding the OMT.**
(DOCX)

**S3 File. Interpretations of low reliability of the integrative explicit power motivation scale.**
(DOCX)

## Author Contributions

**Conceptualization:** Rawan Ghazzawi, Athanasios Chasiotis, Michael Bender, Lina Daouk-Öyry.

**Data curation:** Rawan Ghazzawi.

**Formal analysis:** Rawan Ghazzawi, Athanasios Chasiotis.

**Funding acquisition:** Nicola Baumann.

**Investigation:** Rawan Ghazzawi, Athanasios Chasiotis.

**Methodology:** Rawan Ghazzawi, Michael Bender.

**Project administration:** Rawan Ghazzawi.

**Resources:** Lina Daouk-Öyry, Nicola Baumann.

**Supervision:** Athanasios Chasiotis, Michael Bender, Lina Daouk-Öyry.

**Validation:** Athanasios Chasiotis.

**Writing – original draft:** Rawan Ghazzawi.

**Writing – review & editing:** Rawan Ghazzawi, Michael Bender, Nicola Baumann.

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
