## [Decision Letter · Decision Letter 0]

29 Aug 2023

PONE-D-23-18513Up for the Challenge: Need for Power Drives Employees to Craft Their Jobs and Create Positive Work OutcomesPLOS ONE

Dear Dr. Baumann,

Thank you for submitting your manuscript to PLOS ONE. After careful consideration, we feel that it has merit but does not fully meet PLOS ONE’s publication criteria as it currently stands. Therefore, we invite you to submit a revised version of the manuscript that addresses the points raised during the review process.

We look forward to receiving your revised manuscript.

Kind regards,

Alejandro Vega-Muñoz, Ph.D.

Academic Editor

PLOS ONE

3. Please ensure that you include a title page within your main document. You should list all authors and all affiliations as per our author instructions and clearly indicate the corresponding author.

5. Please include a caption for figure 3.

6. Please upload a copy of Supporting Information materials 1 and 2 which you refer to in your text on pages 11 and 12.

Reviewers' comments:

Reviewer's Responses to Questions

**Comments to the Author**

1. Is the manuscript technically sound, and do the data support the conclusions?

Reviewer #1: Partly

Reviewer #2: Partly

2. Has the statistical analysis been performed appropriately and rigorously? 

Reviewer #1: No

Reviewer #2: Yes

3. Have the authors made all data underlying the findings in their manuscript fully available?

Reviewer #1: Yes

Reviewer #2: Yes

4. Is the manuscript presented in an intelligible fashion and written in standard English?

Reviewer #1: No

Reviewer #2: Yes

5. Review Comments to the Author

Reviewer #1: Thank you for the opportunity to read this manuscript. I think the basic idea is interesting, but there are many aspects that I find problematic, which I will now outline.

Abstract:

I think the abstract can be shortened and make it more accesible.

How are needs and self-attributed needs any different?

In the abstract you report that “Power motives are characterized by the need to have an impact on others” but job crafting is mainly related to the self, hence it is unclear how these motives fit the overall idea of jc.

It is unclear how H4 fits with the overall research question of the study. I do not think you need to specify all the hypotheses in the abstract.

The study design should be described better in the abstract.

You refer to integrative power motive enactment, which is rather unclear in the abstract. Also, from the abstract, it is unclear why you only focused on seeking challenges.

What do you mean by “integrative” power? The use of different terms and acronyms makes it more difficult to follow your paper.

Why should increasing challenges be “one of the most important dimensions of job crafting”?

Intro:

You report that research on job crafting has so far ignored basic motives for power, affiliation and achievement. I do not think this is correct, and I encourage the authors to carefully review JC literature. For example, there are studies considering the link between social norms and job crafting, where social norms are related to the human need for affiliation. A more precise literature review is needed.

Line 55: you report correlation coefficients, but this is not needed here.

Overall, the intro lacks convincing elaboration as to why this study was needed. There are many studies on the role of job crafting for work engagement and satisfaction, so merely stating that this one focuses on nurses is not enough, nor does it refer to the main overall research question of the study. I suggest the authors focus on their main research question and elaborate better as to why this is an important topic to investigate – adopting the research design you adopted – now.

On line 69 you report that job crafting consists of increasing and decreasing challenging job demands. I found this incorrect, as job crafting encompasses increasing challenging demands and decreasing hindering demands, plus optimizing demands, as for current research. Hindering and challenging demands are not the same and shouldn’t be treated as if they were.

Line 74: I was not able to find the variance you report in Rudolph et al.’s paper. This being said I did not find it convincing your argument as to why focusing on seeking challenges only is appropriate, also given that research shows that the different job crafting dimensions function very differently.

Line 101: You report that nursing can be regarded as a power profession. I was not really convinced here about categorizing a profession as defined by a single motive. Could it be that there are different motives driving a profession? Also because of this, I did not find it convincing that power motive is the most relevant driver of job crafting (line 106).

Line 127: You argue that job crafting starts at the cognitive level. However, some more recent research shows that job crafting behaviors seem to anticipate cognitive crafting (e.g., https://doi.org/10.1017/jmo.2022.79 ). Again, a more updated review is needed here to provide a stronger grounding to your hypotheses.

The extensive use of acronyms makes it really difficult to follow your arguments. Please revise using the same terms consistently and avoid acronyms as much as possible.

Method:

How did you recruit participants? What info was given regarding the study aims? In the intro, you referred to the pandemic, but the study was conducted before it, so that argument is not appropriate for the study.

Please detail the procedure used to translate the scales. Please be more specific regarding the money participants received.

In your manuscript, you refer to supplemental material, but I was unable to find it.

Lines 199-206: who coded implicit power motives and what procedure was followed to do so?

I am not familiar with the measure you used to assess seeking challenges (and was not able to see it as supplemental material was not uploaded). However, to my understanding, the items seem to map also cognitive crafting, which is different than behavioral and does not really align with the dimension of seeking challenges.

Results:

It is unclear how moderation was tested.

The reliability coefficient for Explicit Power Motive is below the commonly accepted threshold.

Discussion and conclusion:

Given the issues with the methods and results (importantly, construct validity and reliability), it is unclear what can be concluded from the study.

I am not sure if these aspects can be solved with any revision, as they seem pretty fundamental. Should they succeed in doing so, I encourage the authors to rethink their main research question and revise the paper around it, making sure that the value of this investigation is appropriately presented convincingly.

Reviewer #2: Dear authors, many thanks for giving me the opportunity for reading your paper.

I would suggest you some changes in order to improve its quality.

1. Title: it does not fully inform about the sample characteristics and the relationships tested in the paper. Please, try to better address this point.

2. In the literature review section, you should revise the SPECIFIC research on job crafting among nurses. In the recent time, there is a lot of studies on job crafting with healthcare workers. Please, revise the relevant literature.

3. Some additional explanations about the relationships between job crafting and work engagement should be considered, both in the introduction and in the discussion. For instance, recent studies tested a longitudinal moderated mediation model in which Work Engagement increases over time through an increment in Job Crafting behaviors (Hypothesis 1), while this process is moderated by workers’ age (Hypothesis 2). Studies as Topa and Aranda, 2023 should be discussed as related to your findings.

4. 482 nurses from 18 hospitals means a reduced number of participants from each institution. How many nurses answered the survey in each hospital? Please, better inform about this point due to the potential influence of organizational characteristics on job crafting behaviors.

5. You have a lot of missing data related to the age. Perhaps, your participants do not want to be identified. Please, explain how anonymity has been guaranteed.

6. Job crafting behavior is so complex in nurses working environments, as it is a well regulated work-context. Please, discuss this point.

6. All the discussion section is too limited. Please, compare your findings with other studies among nurses and with the suggestions and comments of the recent meta-analyses and systematic reviews on Job crafting.

6. PLOS authors have the option to publish the peer review history of their article (what does this mean?). If published, this will include your full peer review and any attached files.

Reviewer #1: No

Reviewer #2: No

---

## [Author Response · Author response to Decision Letter 0]

25 Sep 2023

Dear Editor, dear reviewers,

We would like to thank you for the valuable feedback that you have provided us with. We believe that this considerably improved our paper, and we hope that the changes that we made fully address your comments.

Below, we have delineated your comments, provided our corresponding responses, and indicated the locations where revisions have been implemented within the paper. We also highlighted the changes that we made in yellow in the revised manuscript.

We really appreciate the time that you put into revising our paper.

Best regards,

Rawan Ghazzawi on behalf of the co-authors

R2-1 Title: it does not fully inform about the sample characteristics and the relationships tested in the paper. Please, try to better address this point. 

Thank you for your comment. We agree with you and we changed the title accordingly. Pages 1&3

Abstract

R1-1 I think the abstract can be shortened and make it more accesible. 

We agree with you and we shortened the abstract accordingly. Page 2

R1-2 How are needs and self-attributed needs any different? 

We added this sentence to the abstract : “While implicit motives tend to operate at the unconscious, explicit motives operate at the unconscious level.” Page 2

R1-3 In the abstract you report that “Power motives are characterized by the need to have an impact on others” but job crafting is mainly related to the self, hence it is unclear how these motives fit the overall idea of jc. 

We do agree with you that JC is mostly about the self, which is why we are trying to link it to implicit and explicit basic needs. However, our focus here is about the “how” and choosing the power motivation was driven by the idea that we seek to influence our surroundings (prosocially or in a suppressive manner) in order to satisfy our personal need for power, so it is still about the self. But we agree with you that the way we portrayed it might be misleading.

Accordingly, we updated the sentence and now it is : “We focused on power motives, as power is an agentic motive characterized by the need to influence your environment.” Page 2, lines 12-13

R1-4 It is unclear how H4 fits with the overall research question of the study. I do not think you need to specify all the hypotheses in the abstract. 

We removed the hypotheses from the first part of the abstract. When it comes to H4, we added it in order to try to justify as much of the relationships as possible in the study. The two outcomes that we have chosen ( job satisfaction and work engagement) are very relevant to the nursing profession, JC and to each other and it would have made the model weaker if we had not accounted for the relations between them. Although this relationship is not directly related to the RQ, it helps us understand the impact that JC might have on the nurses who engage in it. No change

R1-5 The study design should be described better in the abstract. 

We considered your comment and we added more information to the abstract. Page 2

R1-6 You refer to integrative power motive enactment, which is rather unclear in the abstract. Also, from the abstract, it is unclear why you only focused on seeking challenges. 

We hope that the changes that we made in the abstract make it clearer now.

When it comes to your comment about why we focused on increasing challenging job demands, we added this sentence that we hope clarifies this: 

 “Although power is relevant to job crafting in its entirety, in this study, we link it to increasing challenging job demands due to its relevance to job control, which falls under the umbrella of power.” Page 2, lines 13-15

R1-7 What do you mean by “integrative” power? The use of different terms and acronyms makes it more difficult to follow your paper. 

We mention the word integrative to indicate the level of power we are addressing. At this level, individuals can integrate their emotions (especially negative ones) in a mature manner allowing them to recover after experiencing them and even grow. No change

R1-8 Why should increasing challenges be “one of the most important dimensions of job crafting”? Thank you for your comment. We base this idea on the results of the Meta-analytic confirmatory factor analysis conducted by Rudolph et al (2017), which shows that increasing challenging job demands has a highest loading unto overall JC (.811*) and explaining the largest variance (R2 = .657). No change 

 Introduction 

R1-9 You report that research on job crafting has so far ignored basic motives for power, affiliation and achievement. I do not think this is correct, and I encourage the authors to carefully review JC literature. For example, there are studies considering the link between social norms and job crafting, where social norms are related to the human need for affiliation. A more precise literature review is needed. 

Thank you for mentioning this study. We agree with you that basic needs have already been explored and we are very aware of the literature on that such as: 

Slemp, G. R., & Vella-Brodrick, D. A. (2014). Optimising employee mental health: The relationship between intrinsic need satisfaction, job crafting, and employee well-being. Journal of Happiness Studies, 15, 957-977.

Bipp, T., & Demerouti, E. (2015). Which employees craft their jobs and how? Basic dimensions of personality and employees' job crafting behaviour. Journal of occupational and organizational psychology, 88(4), 631-655.

However, these studies focus only on explicit needs which can only be measured via projective measures and not on implicit needs. Having mentioned that, we thank you for pointing this out to us and for providing us with additional information. Accordingly, we have updated the sentence: 

“Studies on personal antecedents of job crafting have mainly focused on traits such as the big five and self-efficacy (3) and basic needs at the explicit (intrinsic and extrinsic) level (6, 7), which were restricted to self-report. However, implicit (unconscious) basic motives for power, achievement, and affiliation that are driving forces of human behavior (8, 9) have been neglected thus far” Page 3, lines 40-44

R1-10 Line 55: you report correlation coefficients, but this is not needed here. 

We have removed them. Page 4, line 59

R1-11 Overall, the intro lacks convincing elaboration as to why this study was needed. There are many studies on the role of job crafting for work engagement and satisfaction, so merely stating that this one focuses on nurses is not enough, nor does it refer to the main overall research question of the study. I suggest the authors focus on their main research question and elaborate better as to why this is an important topic to investigate – adopting the research design you adopted – now. 

We agree with you that the gap was not very clear in the introduction and accordingly, we added this section:

“however, we still do not know a lot about the personal antecedents of job crafting in general (4) and among nurses specific. This can create a level of unclarity as to who is more likely to engage in job crafting. Addressing this gap is essential as job crafting is a personal and proactive work behavior driven by basic needs (2, 5).” Page 4, lines 34-37

R1-12 Line 74: I was not able to find the variance you report in Rudolph et al.’s paper. This being said I did not find it convincing your argument as to why focusing on seeking challenges only is appropriate, also given that research shows that the different job crafting dimensions function very differently. 

Thank you for your comment. You can find this information on page 122 in 

Rudolph, C. W., Katz, I. M., Lavigne, K. N., & Zacher, H. (2017). Job crafting: A meta-analysis of relationships with individual differences, job characteristics, and work outcomes. Journal of vocational behavior, 102, 112-138.

Although we partially agree with you, we also see the fact that a dimension explains that largest amount of variance in an overall construct is a substantial reason to focus on this dimension because it is also relevant as well to power motivation. Moreover, the fact that the JC dimensions seem to be very different can serve as more support for the need to focus on each JC dimension on its own rather that than on the entire construct of JC. No change 

R1-13 On line 69 you report that job crafting consists of increasing and decreasing challenging job demands. I found this incorrect, as job crafting encompasses increasing challenging demands and decreasing hindering demands, plus optimizing demands, as for current research. Hindering and challenging demands are not the same and shouldn’t be treated as if they were. 

Thank you for your comment. This was a typo and we have corrected it. Page 4, lines 73-74

R1-14 Line 101: You report that nursing can be regarded as a power profession. I was not really convinced here about categorizing a profession as defined by a single motive. Could it be that there are different motives driving a profession? 

Also because of this, I did not find it convincing that power motive is the most relevant driver of job crafting (line 106). 

We agree with you that a single motive does not sum up an entire profession, however, research indicates that nurses chose nursing as a profession driven by a calling to care for and help others, which falls under the umbrella of the power motive. Of course, other motives like affiliation and achievement can also drive the profession, however, given the highly physically and emotionally demanding nature of the nursing profession and the less than ideal employment conditions offered, it is less likely that nurses are driven by the need for achievement and more by the need for power (helping). 

However, we still think that we need to support our argument better and we thank you for pointing this out to us. Accordingly, we have indicated that : 

“Although nurses are driven by different motives when making the decision to join the profession, most of them express that they chose this profession driven by a calling to help and care for others (36).” Page 6, lines 107-109

R1-15 Line 127: You argue that job crafting starts at the cognitive level. However, some more recent research shows that job crafting behaviors seem to anticipate cognitive crafting (e.g., https://doi.org/10.1017/jmo.2022.79 ). Again, a more updated review is needed here to provide a stronger grounding to your hypotheses. 

Thank you for your very helpful comment and for adding the doi! This indeed is a very useful and relevant study. Based on some of the results of this study and building on sensemaking theory, which is highlighted here :

“Behaviors constitute the raw material of cognition, representing what is being made sense of, since people make sense of what they are doing while striving for coherence with their past behaviors (Melo, Dourado, & Andrade, Reference Melo, Dourado and Andrade2021; Weick et al., Reference Weick, Sutcliffe and Obstfeld2005). At the same time, people's cognitive processes are used as a reference to guide future actions, with people making sense of a situation which, in turn, drives following actions (Maitlis & Christianson, Reference Maitlis and Christianson2014).”

We cannot invalidate the idea that JC starts at the cognitive level. Moreover, the results of the study indicated that :

 “employees who reframed their work to highlight the relevance of its perceived resources later reported enriching their jobs and expanding their work roles more often, with benefits for their levels of work engagement over time.” 

However, this was not the case for avoidance crafting, which is understandable. Additionally, we have evidence that shows that job crafting intentions are significantly related to actual job crafting (Tims et al., 2015), which further supports our claim that JC starts at the cognitive level.

Tims, M., Bakker, A. B., & Derks, D. (2015). Job crafting and job performance: A longitudinal study. European Journal of Work and Organizational Psychology, 24(6), 914-928.

In order to address your comment, we have updated the section accordingly: 

“Wrzesniewski and Dutton (5) argue that job crafting starts at the cognitive level and takes place in the workplace where performance indicators serve as expectations to be met. Zhang and Parker (44) have argued that cognitive crafting is relevant in driving job crafting behaviors. Moreover, a recent study by Costantini (45) indicted that employees who cognitively reframed their work engaged in more role expansion.” Page 7, lines 134-138

R1-16 The extensive use of acronyms makes it really difficult to follow your arguments. Please revise using the same terms consistently and avoid acronyms as much as possible. 

We have removed the acronyms in the paper and instead use only implicit power and explicit power. We hope that makes reading the paper clearer. Across the entire paper

R2-2 In the literature review section, you should revise the SPECIFIC research on job crafting among nurses. In the recent time, there is a lot of studies on job crafting with healthcare workers. Please, revise the relevant literature. 

We agree with you that this was not very clear and accordingly we added the following information in the different sections:

“Moreover, a very limited number of these studies investigated the personal antecedents of job crafting in the nursing context (e.g., 17)”

“In the nursing context, this can be volunteering to help at the emergency department when there is a lot of pressure or creating a small support group for junior nurses.”

“Although nurses’ decisions to join the profession are driven by different motives, most of them express that they chose this profession driven by a calling to help and care for others (36).”

“Extensive research supports the relationship between job crafting and work engagement in general (e.g., 2, 27) and among nurses (57).” Page 4, lines 53-54

Page 5, lines 83-84

Page 6, lines 107-109

Page 8, lines 162-163

R2-3 Some additional explanations about the relationships between job crafting and work engagement should be considered, both in the introduction and in the discussion. For instance, recent studies tested a longitudinal moderated mediation model in which Work Engagement increases over time through an increment in Job Crafting behaviors (Hypothesis 1), while this process is moderated by workers’ age (Hypothesis 2). Studies as Topa and Aranda, 2023 should be discussed as related to your findings. 

Thank you for your comment and for including this very interesting study!

We have updated the section accordingly:

“According to the JD-R model, motivated employees engage in job crafting and as result have more resources and experience higher levels of motivation (58). This is referred to as the empirically “gain spiral” that employees create for themselves to increase their work engagement via job crafting (59). However, little is known about the motivational antecedent in this relationship. This is surprising, especially since work engagement is often referred to as a motivational process in the job demands-resources model (60).”

When it comes to the discussion, we have made the following changes as well: 

“We extend this literature by supporting such a robust relationship in an underrepresented non-Western sample and by providing empirical support for the role of motives in increasing the level of motivation among employees (gain spiral) (58).” Pages 8 & 9, lines 163-169

Page 18, lines 365-367

Method

R1-17 How did you recruit participants? What info was given regarding the study aims? In the intro, you referred to the pandemic, but the study was conducted before it, so that argument is not appropriate for the study. In order to address your first two questions, we added the following information:

“Hardcopies of the questionnaires were distributed among the nurses in each of the participating hospitals and the nurses were requested to return the filled questionnaires. At the beginning of the study the participants had the chance to read about the aim of the study and provide their consent if applicable.”

When it comes to your last comment about the COVID-19 argument in the introduction, thank

---

## [Decision Letter · Decision Letter 1]

5 Apr 2024

PONE-D-23-18513R1Up for the Challenge: Power Motive Congruence Drives Nurses to Craft Their Jobs and Experience Well-beingPLOS ONE

Dear Dr. Baumann,

Thank you for submitting your manuscript to PLOS ONE. After careful consideration, we feel that it has merit but does not fully meet PLOS ONE’s publication criteria as it currently stands. Therefore, we invite you to submit a revised version of the manuscript that addresses the points raised during the review process.

The following reviewers’ suggestions seem to be improved or clarified for further revision.

1. Concerns about Conceptual Justifications and Theoretical Framework:

• The revision is appreciated, but not all previous concerns were adequately addressed.

• The justification for focusing solely on seeking challenges in job crafting needs further explanation, as the choice does not seem to align with the strong theoretical support for the relationship between job demands and employee control.

• There is a need to clarify why other dimensions of job crafting were overlooked despite the specific request in the previous report.

• Misrepresentation of sources in the manuscript, such as linking explicit power to job crafting without relevant references, raises concerns about theoretical grounding.

2. Issues with Methodological Approach:

• The nested structure of the data should have prompted consideration of multilevel analyses, which appears to have been overlooked.

• Lack of clarity on how moderation was tested, including the handling of predictors and the impact of adding interaction effects.

• Concerns regarding the construct validity of the newly developed scale and its alignment with job crafting behaviors versus motives.

• Uncertainty around how operant motive tests were coded and treated as a continuous variable despite the scale's inherent categorical nature, impacting reliability.

3. Construct Validity and Scale Development:

• Doubts raised about whether the scale measures job crafting behaviors or motives, emphasizing the need for clarity and validation.

• Specific item-level critiques, questioning the presence of instrumental motives in certain scale items and the interpretation of performance-related tasks within job crafting.

4. Theoretical Coherence and Discussion Alignment:

• Issues highlighted in the discussion section regarding the alignment with job crafting concepts, particularly the emphasis on successfully completing challenging tasks versus proactively seeking challenges.

• The introduction of the expectancy theory without prior mention or integration into the manuscript underscores the need for a more coherent theoretical framework.

We look forward to receiving your revised manuscript.

Kind regards,

Chen-Wei Yang

Academic Editor

PLOS ONE

Additional Editor Comments:

PONE-D-23-18513

The paper explores how power motives, both implicit and explicit, drive employees to engage in job crafting by increasing challenging job demands, ultimately impacting work-related outcomes. Owing few studies have explored it in the context of nursing and previous studies were restricted to self-report, the research employed a questionnaire survey to collect data from 482 nurses from 18 hospitals across Lebanon. The results indicate that explicit and implicit power motives are crucial for promoting job crafting, which in turn leads to positive work outcomes like work engagement and job satisfaction. However, the following reviewers’ suggestions seem to be improved or clarified for further revision.

1. Concerns about Conceptual Justifications and Theoretical Framework:

• The revision is appreciated, but not all previous concerns were adequately addressed.

• The justification for focusing solely on seeking challenges in job crafting needs further explanation, as the choice does not seem to align with the strong theoretical support for the relationship between job demands and employee control.

• There is a need to clarify why other dimensions of job crafting were overlooked despite the specific request in the previous report.

• Misrepresentation of sources in the manuscript, such as linking explicit power to job crafting without relevant references, raises concerns about theoretical grounding.

2. Issues with Methodological Approach:

• The nested structure of the data should have prompted consideration of multilevel analyses, which appears to have been overlooked.

• Lack of clarity on how moderation was tested, including the handling of predictors and the impact of adding interaction effects.

• Concerns regarding the construct validity of the newly developed scale and its alignment with job crafting behaviors versus motives.

• Uncertainty around how operant motive tests were coded and treated as a continuous variable despite the scale's inherent categorical nature, impacting reliability.

3. Construct Validity and Scale Development:

• Doubts raised about whether the scale measures job crafting behaviors or motives, emphasizing the need for clarity and validation.

• Specific item-level critiques, questioning the presence of instrumental motives in certain scale items and the interpretation of performance-related tasks within job crafting.

4. Theoretical Coherence and Discussion Alignment:

• Issues highlighted in the discussion section regarding the alignment with job crafting concepts, particularly the emphasis on successfully completing challenging tasks versus proactively seeking challenges.

• The introduction of the expectancy theory without prior mention or integration into the manuscript underscores the need for a more coherent theoretical framework.

Reviewers' comments:

Reviewer's Responses to Questions

**Comments to the Author**

1. If the authors have adequately addressed your comments raised in a previous round of review and you feel that this manuscript is now acceptable for publication, you may indicate that here to bypass the “Comments to the Author” section, enter your conflict of interest statement in the “Confidential to Editor” section, and submit your "Accept" recommendation.

Reviewer #1: (No Response)

Reviewer #3: (No Response)

2. Is the manuscript technically sound, and do the data support the conclusions?

Reviewer #1: No

Reviewer #3: Yes

3. Has the statistical analysis been performed appropriately and rigorously? 

Reviewer #1: No

Reviewer #3: Yes

4. Have the authors made all data underlying the findings in their manuscript fully available?

Reviewer #1: Yes

Reviewer #3: Yes

5. Is the manuscript presented in an intelligible fashion and written in standard English?

Reviewer #1: Yes

Reviewer #3: Yes

6. Review Comments to the Author

Reviewer #1: Dear authors,

thank you for revising the manuscript.

While I appreciate this updated version of the manuscript more than the previous one, I do not think that all my previous concerns were properly considered and addressed. Moreover, given the additional information available now in the supplementary material, I have some serious concerns about the validity of the measures used, outlined below.

Intro:

I do not follow how the fact that the relation between job demands and employee control is strongly supported by the theory justifies your choice of focusing on seeking challenges only (lines 99-100).

Related to this, I’d like to point out that in my previous report, I asked specifically to justify why focusing on seeking challenges only may be appropriate, I never stated you should have considered all job crafting dimensions together. My question was why you only focused on seeking challenges and ignored the other job crafting dimensions. This question remains.

On line 168 you report that job crafting is a planned behavior predicted by high explicit power. However, the references you used to back this statement are not about job crafting at all.

Method:

It seems that your data has a nested structure, so you should have at least checked whether multilevel analyses are appropriate to test your hypotheses.

When I asked about how moderation was tested, I meant that you should have specified whether you centered your predictors and how the model changed when adding the interaction effect. I know what it means that there is a moderation effect.

Now that the scale is available in the supplemental material, I have more doubts about its construct validity, whether it maps at all job crafting in the form of seeking challenges or whether the constructs that you measure are something very different, both in comparison to job crafting and in the two dimensions proposed, i.e., affective and instrumental motives. The development of a new scale requires checking whether this relates to the original construct, as well as data to check its validity, which is missing here. Moreover, the procedures used to develop this new scale are rather unclear. I think your scale does not measure job crafting behaviors but rather the motives to engage in job crafting, which is something very different and requires a deeper investigation to understand if this scale you developed is actually valid.

As another issue, (so please, in case you are invited to resubmit a revised version, I’d appreciate a response to each of my comments), while in most of the items, I can somehow see how instrumental considerations are expressed, I do not see this in the item “I try to understand how my work tasks are related to one another”. Where is the instrumental motive here? Similarly, I found it unclear whether the item “I take on extra tasks because this is part of my performance appraisal” actually maps some instrumental motive that comes from within the employee rather than as something that is expected – I would assume that if something is part of your performance appraisal you don’t really have much choice about whether to do it or not.

Similarly to the scale developed to measure affective and instrumental job crafting motives, I have some serious concerns regarding the way you coded operant motive tests. From what I read in the supplement file, it seems that you treated it as a continuous variable, but then the scores from 1 to 4 refer to approach crafting and 5 refers to avoidance crafting, i.e., something completely different. Yet, you treat it all together as a single continuous variable. This is a serious concern, and I think can partially explain the issues with the unacceptable reliability.

Minor:

In the discussion, you start by saying that successfully completing challenging job demands is important. Yet, job crafting is not about that, it’s about proactively looking for more challenging job demands.

On line 406 you cite the expectancy theory of motivation, but this comes across as rather out of the blue, given that it wasn’t mentioned in the intro nor anywhere in the manuscript. This further highlights the lack of a coherent theoretical framework underlying this study.

Line 411-413 – I do not understand how reference 83 is relevant to your study findings.

The theoretical contributions are not well grounded in the existing literature. It is unclear how they specifically advance what we already know.

Reviewer #3: �The study is interesting and scientifically sounds, the methodological approach is clear and can be replicated, using all or few dimensions like focusing only on job crafting and work engagement.

In the introduction it will be more advised to include information about the nature & current situation of job crafting in healthcare settings in Lebanon (how common is it among researchers in the same region), which is usually governed by ministries or authorities at national level. And in order to bridge the gap and implement the recommendations, policy makers (specifically human resources managers) need to be convinced about it.

Usually, this topic will be combined by assessment of burnout, which was missing, using Maslach Burnout Inventory as example. The MBI had a section about personal achievement that can be considered as part of positive outcomes of job crafting.

Line 69, a contradicting statement (implicit motives require objective techniques because they are not consciously accessible to individuals)

Lack comparison with other professions, in reference to the nature of healthcare settings and presence of multidisciplinary approach.

I found this article is worthy to be published.

7. PLOS authors have the option to publish the peer review history of their article (what does this mean?). If published, this will include your full peer review and any attached files.

Reviewer #1: No

Reviewer #3: **Yes: **Dr Al Ounoud Mohamed Al Marzouqi

---

## [Author Response · Author response to Decision Letter 1]

20 May 2024

Dear Editor, dear reviewers,

We would like to thank you for the valuable feedback that you have provided us with. We believe that this considerably improved our paper, and we hope that the changes that we made fully address your comments.

Below, we have delineated your comments, provided our corresponding responses, and indicated the locations where revisions have been implemented within the paper. We also highlighted the changes that we made in yellow (deleted in red) in the revised manuscript.

We really appreciate the time that you put into revising our paper.

Best regards,

Rawan Ghazzawi on behalf of the co-authors

E1.1 The revision is appreciated, but not all previous concerns were adequately addressed. 

Author's Response (AR): We hope that we have addressed all your concerns with the changes that we have made. 

E1.2 The justification for focusing solely on seeking challenges in job crafting needs further explanation, as the choice does not seem to align with the strong theoretical support for the relationship between job demands and employee control. 

AR: We have added more justification in order to support our choice to focus on increasing challenging job demands. 

Lines changed: Lines 99-110

E1.3 There is a need to clarify why other dimensions of job crafting were overlooked despite the specific request in the previous report. 

AR: We hope that the changes that we have made address this comment. Lines 99-110

E1.4 Misrepresentation of sources in the manuscript, such as linking explicit power to job crafting without relevant references, raises concerns about theoretical grounding 

AR: We have addressed this issue whenever it was pointed out.

E1.5 The nested structure of the data should have prompted consideration of multilevel analyses, which appears to have been overlooked. 

AR: We appreciate you pointing this out to us, thank you! We provided more information about this in our response below (R1.4) which we hope now addresses this issue. We also conducted an additional analysis to calculate the ICC scores and support of choice to analyze the variables at the individual level.

Lines changed: Lines 288-294

E1.6 Lack of clarity on how moderation was tested, including the handling of predictors and the impact of adding interaction effects. 

AR: We have indicated that we centered the explicit motives variable and not the implicit motives one because the latter is a binary one. Moreover, added some information on how much variance was explained when the moderation was added. 

Lines changed: Lines 329-330

E1.7 Concerns regarding the construct validity of the newly developed scale and its alignment with job crafting behaviors versus motives.

AR: We have addressed this comment by including all the information about the scale development process that we had decided to exclude before for the purpose of conciseness. We hope that this addresses your comment in its entirety, however, if not, we would be happy to elaborate further.

Lines changed: 269-272 And Supplemental material (S1.a)

E1.8 Uncertainty around how operant motive tests were coded and treated as a continuous variable despite the scale's inherent categorical nature, impacting reliability.

AR: We used it as a binary/ categorical variable as was highlighted in lines 305-306, which is consistent with previous motive research (Baumann & Scheffer, 2010). We hope that this addresses the issue. 

E1.9 Doubts raised about whether the scale measures job crafting behaviors or motives, emphasizing the need for clarity and validation. 

AR: Kindly refer to E1.7 

E1.10 Specific item-level critiques, questioning the presence of instrumental motives in certain scale items and the interpretation of performance-related tasks within job crafting 

AR: We hope that we addressed the reviewer’s concerns about the specific items in our response to R1.7

E1.11 Issues highlighted in the discussion section regarding the alignment with job crafting concepts, particularly the emphasis on successfully completing challenging tasks versus proactively seeking challenges.

AR: This was adjusted.

E1.12 The introduction of the expectancy theory without prior mention or integration into the manuscript underscores the need for a more coherent theoretical framework. 

AR: This was removed.

R1 Introduction

R1.1 I do not follow how the fact that the relation between job demands and employee control is strongly supported by the theory justifies your choice of focusing on seeking challenges only (lines 99-100). 

AR: We have added more justification in order to support our choice to focus on increasing challenging job demands. 

Lines changed: Lines 92-101

R1.2 Related to this, I’d like to point out that in my previous report, I asked specifically to justify why focusing on seeking challenges only may be appropriate, I never stated you should have considered all job crafting dimensions together. My question was why you only focused on seeking challenges and ignored the other job crafting dimensions. This question remains.

AR: We apologize for not properly addressing your previous question, this was not our intention. We hope that we have now addressed it by the changes that we have made. 

We decided to focus on this dimension because it is the one that is most likely to be influenced by this level of power. Taking on challenges at work reflects a level of maturity, which is expressed at the third level of the power motive. Individuals at this level can deal with negative emotions should they arise, which makes them better equipped to tackle challenging job demands because they know that they can finish them. At this level, you do not avoid demands, because you are empowered by positive affect that encourages you to approach tasks at work rather than avoid them. 

Lines changed: Lines 92-101

R1.3 On line 168 you report that job crafting is a planned behavior predicted by high explicit power. However, the references you used to back this statement are not about job crafting at all. 

AR: Thank you for pointing this out to us. References (1,5) are about job crafting, however, the references that come after them (48,49) are indeed not about job crafting. However, they were used to explain the motive congruence phenomenon in general. We hope that the changes that we made address you concern. 

Lines changed: Lines 166-169

Method: 

R1.4 It seems that your data has a nested structure, so you should have at least checked whether multilevel analyses are appropriate to test your hypotheses.

AR: We appreciate you pointing this out to us, thank you! We did indeed consider the multilevel, however considering that our data is all at the individual level and indicates individual-level variables (job crafting, implicit and explicit motives, job satisfaction, and work engagement), it was hard for us to use multi-level analysis. 

None of the variables that we have collected indicated variability in terms of organizational or team-level behavior. Even if we aggregated the variables to create an organizational-level variable, we would have done that based on no theoretical grounding. 

However, we did conduct an additional analysis to calculate the ICC scores and support of choice to analyze the variables at the individual level.

Your comment made us consider this design for the next study, and for that, we will collect data from multiple sources (while making sure we have enough members from each group).

Lines changed: Lines 288-294

R1.5 When I asked about how moderation was tested, I meant that you should have specified whether you centered your predictors and how the model changed when adding the interaction effect. I know what it means that there is a moderation effect. 

AR: Apologies if it came off as us explaining what the moderation effect means, this was not our intention! 

We have indicated that we centered the explicit motives variable and not the implicit motives one because the latter is a binary one. I hope this addresses your comment now.

When it comes to the other part of your comment how the model changed when adding the interaction effect, we have now added information about the R-squared before and after including the moderation variables. I hope this addresses your comment, but if not, we are willing to add more information. 

Lines changed: Lines 305-306 And Lines 329-330

R1.6 Now that the scale is available in the supplemental material, I have more doubts about its construct validity, whether it maps at all job crafting in the form of seeking challenges or whether the constructs that you measure are something very different, both in comparison to job crafting and in the two dimensions proposed, i.e., affective and instrumental motives. 

The development of a new scale requires checking whether this relates to the original construct, as well as data to check its validity, which is missing here. Moreover, the procedures used to develop this new scale are rather unclear. I think your scale does not measure job crafting behaviors but rather the motives to engage in job crafting, which is something very different and requires a deeper investigation to understand if this scale you developed is actually valid.

AR: Thank you for your comment, and we agree with you. We have indeed piloted this new scale but decided to exclude this step for simplification and word count. In hindsight, maybe it was not the best decision because it made the process less clear. We have now included this information in the supplemental material. This information shows the different validity tests that we conducted to make sure that we have a valid scale.

When it comes to the other part of your comment, we delineated the original job crafting scale (increasing challenging job demands) by Tims et al. (2012) into instrumental versus effective with an aim to relate it better to explicit and implicit motives. However, as is clear in the pilot study (that is now in the supplemental material) the two sets of items behaved similarly and loaded unto the same major dimensions. Therefore, we ended up using the two set of items as one scale that used the same items as the ones in the original scale by Tims et al (2012), but that added a motivational element to them. We also reported the CFA of ONE job crafting scale instead of two, which shows that we have used it as one scale in the analysis. Based on your comment, and in order to avoid any confusion, we made this clear under the job crafting section under measures and we hope that the information about the validation process that we added also makes it clearer.

When it comes to your comment about the scale itself, and while we do see your point (measuring motives instead of behaviors), we think that it still measures actual behavior. We think so for two main reasons: the first is the fact that we adapted these items from the original scale that measures JC behavior by Tims et al (2012), while not diverting a lot in terms of wording. We just kept the behavior and added the motivational element to it. The second reason is a more technical one as this adapted JC scale behaved in a way similar to the original scale (positively related to proactive personality and personal initiative and negatively related to cynicism). 

We hope that this addresses your comment in its entirety, however, if not, we would be happy to elaborate further. 

Lines changed: Lines 269-272 And Supplemental material (S1.a)

R1.7 As another issue, (so please, in case you are invited to resubmit a revised version, I’d appreciate a response to each of my comments), while in most of the items, I can somehow see how instrumental considerations are expressed, I do not see this in the item “I try to understand how my work tasks are related to one another”. Where is the instrumental motive here? Similarly, I found it unclear whether the item “I take on extra tasks because this is part of my performance appraisal” actually maps some instrumental motive that comes from within the employee rather than as something that is expected – I would assume that if something is part of your performance appraisal you don’t really have much choice about whether to do it or not.

AR: We see your point when it comes to this item. However, before we provide you with our reasoning, we would like to point out that we tried our best to stay as close as possible to the original scale. This meant that we did not change the items that, based on face value, looked instrumental. Having mentioned that, our reasoning for adapting this item in such a way was to highlight that employees engage in job crafting in order to “understand” how their work tasks are related to one another so that they can work better instead of enjoying this process, which we thought would be more affective. 

When it comes to the second item that you have highlighted, and again just as the case with a lot of points that you bring forth, we see what you mean and we agree with it. However, we think that taking extra tasks because you will avoid punishments/ receive benefits can be considered instrumental. This was also our way to make sure that there is a clear difference between what you are doing because you have to versus because you enjoy it.

We hope that this addresses your comment in its entirety, however, if not, we would be happy to elaborate further 

R1.8 Similarly to the scale developed to measure affective and instrumental job crafting motives, I have some serious concerns regarding the way you coded operant motive tests. From what I read in the supplement file, it seems that you treated it as a continuous variable, but then the scores from 1 to 4 refer to approach crafting and 5 refers to avoidance crafting, i.e., something completely different. Yet, you treat it all together as a single continuous variable. This is a serious concern, and I think can partially explain the issues with the unacceptable reliability. 

AR: In the supplemental file, we described how coders differentiate the five levels of motive enactment. Even if aggregating scores across all five levels is the classical way to measure the general power motive, we were specifically interested in the integrative power motive (level 3) in this study. Thus, we used only level 3, that is, the integrative motive enactment. Nevertheless, you raise an excellent point and this would have been an issue had we considered all five levels together as a continuous variable. However, we used only level 3 as a binary/ categorical variable as was highlighted in lines 311-312. We hope that this addresses the issue. 

R1.9 In the discussion, you start by saying that successfully completing challenging job demands is important. Yet, job crafting is not about that, it’s about proactively looking for more challenging job demands. 

AR: Thank you for pointing this out to us; we agree with you. We have changed the discussion section. 

Lines changed: Lines 357-358

R1.10 On line 406 you cite the expectancy theory of motivation, but this comes across as rather out of the blue, given that it wasn’t mentioned in the intro nor anywhere in the manuscript. This further highlights the lack of a coherent theoretical framework underlying this study. 

AR: We see your point and accordingly, we have removed the expectancy theory of motivation to avoid the impression of lacking coherence. 

Lines changed: Lines 414-419 Highlighted in red

R1.11 Line 411-413 – I do not understand how reference 83 is relevant to your study findings.

AR: We have included this study because we thought that it is in line with the findings of our study: employees who increase their challenging job demands and who have high personal resources (which is motive congruence in our case) experience higher levels of work engagement. We have added another sentence that articulates how we think our study links to the Bakker and Sanz-Vergel one. 

Lines changed: Lines 419-423

R1.12 The theoretical contributions are not well grounded in the existing literature. It is unclear how they specifically advance what we already know. 

AR: Thank you for your comment. We hope that the changes that we made have addressed your comment properly. 

Lines changed: Lines 432-437

R2 

R2.1 In the intro

---

## [Decision Letter · Decision Letter 2]

3 Jul 2024

PONE-D-23-18513R2Up for the Challenge: Power Motive Congruence Drives Nurses to Craft Their Jobs and Experience Well-beingPLOS ONE

Dear Dr. Baumann,

Thank you for submitting your manuscript to PLOS ONE. After careful consideration, we feel that it has merit but does not fully meet PLOS ONE’s publication criteria as it currently stands. Therefore, we invite you to submit a revised version of the manuscript that addresses the points raised during the review process.

We look forward to receiving your revised manuscript.

Kind regards,

Chen-Wei Yang

Academic Editor

PLOS ONE

**Additional Editor Comments:**

The reviewers have provided detailed feedback on the manuscript titled "Up for the Challenge: Power Motive Congruence Drives Nurses to Craft Their Jobs and Experience Well-being." Their comments highlight several areas for improvement, focusing on the abstract, introduction, methodology, results, and overall conclusions.

Abstract:

The abstract needs to be shortened and made more accessible to a wider audience. It should clarify the difference between needs and self-attributed needs, and explain how power motives fit within the context of job crafting, which typically relates to the self. Additionally, the study design should be better described without specifying all hypotheses. Terms like “integrative power motive enactment” are unclear and need explanation, particularly the focus on seeking challenges. It is also essential to justify why increasing challenges is considered a critical dimension of job crafting.

Introduction:

The introduction should provide a more comprehensive review of the job crafting literature, acknowledging that previous research has addressed basic motives such as power, affiliation, and achievement. Unnecessary details, such as correlation coefficients, should be removed. A stronger rationale for the study is needed, focusing on the main research question and its importance. The description of job crafting should accurately distinguish between increasing challenging demands and decreasing hindering demands. References and variance reports must be accurate, and the argument that nursing is driven primarily by power motives should be reconsidered, exploring other potential motives. Recent research on job crafting behaviors and cognitive crafting should be included, and the extensive use of acronyms should be reduced for better clarity and consistency.

Methodology:

Details on participant recruitment, the information provided regarding study aims, and translation procedures for scales need to be specified. Participant compensation details should also be included. It is crucial to ensure that all supplemental material is accessible. The process and personnel involved in coding implicit power motives should be clearly outlined. The measure used to assess seeking challenges should align with behavioral job crafting rather than cognitive crafting.

Results and Analysis:

The moderation testing process requires clarification. The reliability of the Explicit Power Motive measure is below the commonly accepted threshold and needs to be addressed. Additionally, the number of statistical tests should be reduced to maintain clarity and impact.

Discussion and Conclusion:

Given the issues with methods and results, particularly regarding construct validity and reliability, the study's conclusions are unclear. Concerns about the originality of the research, ambiguity in findings, and limited generalizability need to be addressed.

Reviewers' comments:

Reviewer's Responses to Questions

**Comments to the Author**

1. If the authors have adequately addressed your comments raised in a previous round of review and you feel that this manuscript is now acceptable for publication, you may indicate that here to bypass the “Comments to the Author” section, enter your conflict of interest statement in the “Confidential to Editor” section, and submit your "Accept" recommendation.

Reviewer #4: All comments have been addressed

Reviewer #5: All comments have been addressed

2. Is the manuscript technically sound, and do the data support the conclusions?

Reviewer #4: Yes

Reviewer #5: Yes

3. Has the statistical analysis been performed appropriately and rigorously? 

Reviewer #4: Yes

Reviewer #5: Yes

4. Have the authors made all data underlying the findings in their manuscript fully available?

Reviewer #4: Yes

Reviewer #5: No

5. Is the manuscript presented in an intelligible fashion and written in standard English?

Reviewer #4: Yes

Reviewer #5: Yes

6. Review Comments to the Author

Reviewer #4: Author(s) all of comments and suggestions have been thoroughly addressed in the revised manuscript. Now this paper is insightful and enhancing the quality of our work, and ensure that the paper meets the highest standards of academic rigor.

Reviewer #5: kindly check your grammar through the entire work. As this will improve the clarity of the manuscript.

7. PLOS authors have the option to publish the peer review history of their article (what does this mean?). If published, this will include your full peer review and any attached files.

Reviewer #4: No

Reviewer #5: No

---

## [Author Response · Author response to Decision Letter 2]

19 Aug 2024

Dear Editor, 

We would like to thank you for the valuable feedback that you have provided us with. We believe that this considerably improved our paper, and we hope that the changes that we made fully address your comments.

Below, we have delineated your comments (numbered), provided our corresponding responses, and indicated the locations where revisions have been implemented within the paper. We also highlighted the changes that we made in yellow (deleted in red) in the revised manuscript.

We really appreciate the time that you put into revising our paper.

Best regards,

Rawan Ghazzawi on behalf of the co-authors

Comment (C): 1.1 The abstract needs to be shortened and made more accessible to a wider audience. 

Response (R): Thank you for this comment. We shortened the abstract and filtered out the terms that were too technical and could be mentioned at a later stage throughout the manuscript. Lines 24-37

C: 1.2 It should clarify the difference between needs and self-attributed needs, and explain how power motives fit within the context of job crafting, which typically relates to the self. 

R: This was not mentioned in the abstract, however, we made sure to clarify the difference the first time we mentioned these terms in the manuscript. Line 119

C: 1.3 Additionally, the study design should be better described without specifying all hypotheses. 

R: We added more information about the study design and removed the hypotheses. Lines 28-36

C: 1.4 Terms like “integrative power motive enactment” are unclear and need explanation, particularly the focus on seeking challenges. 

R: This term was not used throughout the paper, however, we added information in the text that we hope makes this link clearer. Lines 139-142

C: 1.5 It is also essential to justify why increasing challenges is considered a critical dimension of job crafting. 

R: We have added information in the abstract that we hope addresses this comment. Lines 31-32

C: 2.1 The introduction should provide a more comprehensive review of the job crafting literature, acknowledging that previous research has addressed basic motives such as power, affiliation, and achievement.. 

R: Given that the research on the link between job crafting and basic needs/motives is very limited, we made sure to include all of the studies that have addressed this relationship. However, we think that adding more information can indeed strengthen our introduction and accordingly we have added more details in the second paragraph of the introduction that we hope addresses your comment. Lines 57-60

C: 2.2 Unnecessary details, such as correlation coefficients, should be removed. 

R: We have removed this information. Line 104

C: 2.3 A stronger rationale for the study is needed, focusing on the main research question and its importance. 

R: We agree with you that the main research question was not clearly indicated, and accordingly, we have added one we hope makes the main research aim of the study clearer. Lines 77-79

C: 2.4 The description of job crafting should accurately distinguish between increasing challenging demands and decreasing hindering demands. 

R: We have added more information about the difference between increasing challenging job demands and decreasing hindering job demands. Lines 93-98

C: 2.5 References and variance reports must be accurate, 

R: Thank you for indicating that, we have reviewed the paper to make sure that this information is accurate. Throughout the paper.

C: 2.6 the argument that nursing is driven primarily by power motives should be reconsidered, exploring other potential motives 

R: We hope that the changes that we have made make this argument more convincing. Line 136

C: 2.7 Recent research on job crafting behaviors and cognitive crafting should be included 

R: Thank you for this suggestion, however, we think that cognitive crafting is beyond the scope of our study and adding it will add even more concepts and distract the reader. No change.

C: 2.8 the extensive use of acronyms should be reduced for better clarity and consistency 

R: Thank you for this comment, however, the previous versions of the paper did include a lot of acronyms and based on the comments that we received, we reduced them. Based on your comment, we reviewed the paper again and made sure there are no unnecessary acronyms. Throughout the paper.

C: 3.1 Methodology:

Details on participant recruitment, the information provided regarding study aims, and translation procedures for scales need to be specified. We have added more information about participant recruitment under the sample and procedure section. In the current version of the study, we have included information about how the scales were translated “Questionnaires were administered in Arabic after careful scale adaptation processes [72, 73]. A bilingual research team translated and back-translated the scales. A university professor in Arabic reviewed the items, and a committee of bilingual researchers finalized the scales. Cognitive interviews with a small group of nurses assessed item clarity.” 

R: We hope that this information addresses your comment. Lines 228-229

C: 3.2 Participant compensation details should also be included. 

R: We have added that participants were not compensated for participating in the study. Lines 236-237

C: 3.3 It is crucial to ensure that all supplemental material is accessible. The process and personnel involved in coding implicit power motives should be clearly outlined. 

R: We have indeed provided information about the coding process in the supplemental material: “Coding the answers was conducted by the first author who is bilingual (English, Arabic), and underwent coding training with the second author until the percentage of agreement between the trainer and the trainee reached more than 80%.” 

We hope that this information addresses your comment. Supplemental material S1.b.

C: 3.4 The measure used to assess seeking challenges should align with behavioral job crafting rather than cognitive crafting. 

R: We are glad that you think so because this is exactly what we aim to measure. Cognitive crafting is beyond the scope of your study and was not measured as part of the research that we have conducted. No change.

C: 4.1 Results and Analysis:

R: The moderation testing process requires clarification. We have added information about the variables that we included in the moderation analysis. We hope that this makes the moderation analysis clearer. Lines 339-340

C: 4.2 The reliability of the Explicit Power Motive measure is below the commonly accepted threshold and needs to be addressed. 

R: This is very true and accordingly, we have included a section in the supplemental material that addresses this issue. We also refer to this section in the paper. Supplemental Material S1.c.

C: 4.3 Additionally, the number of statistical tests should be reduced to maintain clarity and impact. 

R: Thank you for this suggestion, however, we do not know which statistical tests we should remove without impacting the transparency of our analysis. We would appreciate it if you can provide us with the specific tests that you want to be removed so that we can move them to the supplemental material. No change.

C: 5.1 Discussion and Conclusion:

Given the issues with methods and results, particularly regarding construct validity and reliability, the study's conclusions are unclear 

R: We hope that the changes that we made based on your comments have addressed these issues. We really appreciate your help and support throughout this process and hope that the current version of the paper has clearer conclusions. Throughout the paper.

C: 5.2 Concerns about the originality of the research, ambiguity in findings, and limited generalizability need to be addressed. 

R: When it comes to the originality of the research, this study is the first to link implicit motives to job crafting, which makes it very original. Based on your comment, we have strengthened the research gap mentioned in the introduction, so we hope that this addresses your concern. Moreover, we have included limitations related to the findings and the generalizability of the study under the “Limitations and Future Research Directions” section that we hope addresses your concerns. Lines 475-502

---

## [Editor Report · Decision Letter 3]

5 Sep 2024

Up for the Challenge: Power Motive Congruence Drives Nurses to Craft Their Jobs and Experience Well-being

PONE-D-23-18513R3

Dear Dr. Baumann:

We’re pleased to inform you that your manuscript has been judged scientifically suitable for publication and will be formally accepted for publication once it meets all outstanding technical requirements.

Kind regards,

Chen-Wei Yang

Academic Editor

PLOS ONE

Additional Editor Comments (optional):

Based on the author's thorough responses to the reviewers' comments and the significant revisions made to the manuscript, I believe the paper is now suitable for publication.
---

## [Editor Report · Acceptance letter]

24 Sep 2024

PONE-D-23-18513R3 

PLOS ONE

Dear Dr. Baumann, 

I'm pleased to inform you that your manuscript has been deemed suitable for publication in PLOS ONE. Congratulations! Your manuscript is now being handed over to our production team.

Kind regards, 

on behalf of

Professor Chen-Wei Yang 

Academic Editor

PLOS ONE